# Changes in elbow flexion EMG morphology during adjustment of deep brain stimulator in advanced Parkinson's disease

**Verneri Ruonala**[1] *, **Eero Pekkonen**[2,3], **Olavi Airaksinen**[4], **Markku Kankaanpää**[5], **Pasi A. Karjalainen**[1], **Saara M. Rissanen**[1]

**1** Department of Applied Physics, University of Eastern Finland, Kuopio, Finland, **2** Department of Neurology, Helsinki University Hospital, Helsinki, Finland, **3** Department of Clinical Neurosciences (Neurology), University of Helsinki, Helsinki, Finland, **4** Department of Physiology and Rehabilitation Medicine, Kuopio University Hospital, Finland, **5** Department of Physiology and Rehabilitation Medicine, Tampere University Hospital, Tampere, Finland

* verneri.ruonala@uef.fi

**Data Availability Statement:** The data required to reproduce the results of the study has been included as a Supporting information file.

## Abstract

### Objective

Deep brain stimulation (DBS) is an effective treatment for motor symptoms of advanced Parkinson's disease (PD). Currently, DBS programming outcome is based on a clinical assessment. In an optimal situation, an objectively measurable feature would assist the operator to select the appropriate settings for DBS. Surface electromyographic (EMG) measurements have been used to characterise the motor symptoms of PD with good results; with proper methodology, these measurements could be used as an aid to program DBS.

### Methods

Muscle activation measurements were performed for 13 patients who had advanced PD and were treated with DBS. The DBS pulse voltage, frequency, and width were changed during the measurements. The measured EMG signals were analysed with parameters that characterise the EMG signal morphology, and the results were compared to the clinical outcome of the adjustment.

### Results

The EMG signal correlation dimension, recurrence rate, and kurtosis changed significantly when the DBS settings were changed. DBS adjustment affected the signal recurrence rate the most. Relative to the optimal settings, increased recurrence rates (median ± IQR) 1.1 ± 0.5 (−0.3 V), 1.3 ± 1.1 (+0.3 V), 1.7 ± 0.4 (−30 Hz), 1.7 ± 0.8 (+30 Hz), 2.0 ± 1.7 (+30 μs), and 1.5 ± 1.1 (DBS off) were observed. With optimal stimulation settings, the patients' Unified Parkinson's Disease Rating Scale motor part (UPDRS-III) score decreased by 35% on average compared to turning the device off. However, the changes in UPRDS-III arm tremor and rigidity scores did not differ significantly in any settings compared to the optimal stimulation settings.

**Funding:** This study was supported by the Academy of Finland under project 252748. The funders had no role in study design, data collection and analysis, decision to publish, or preparation of the manuscript.

**Competing interests:** The authors have read the journal's policy and have the following competing interests: Verneri Ruonala is an inventor in the patent applications PCT/FI2019/050163 and WO2020174122A1. Saara M. Rissanen and Pasi A. Karjalainen are inventors in the patent applications EP18159445.8, PCT/EP2019/055002, PCT/FI2019/050163, WO2020174122A1, and WO2019166557A1; they are also co-funders of the company Adamant Health Ltd., which develops software for surface electromyography analysis. This does not alter our adherence to PLOS ONE policies on sharing data and materials.

## Conclusion

Adjustment of DBS treatment alters the muscle activation patterns in PD patients. The changes in the muscle activation patterns can be observed with EMG, and the parameters calculated from the signals differ between optimal and non-optimal settings of DBS. This provides a possibility for using the EMG-based measurement to aid the clinicians to adjust the DBS.

## Introduction

Parkinson's disease (PD) is a progressive neurodegenerative disease that has an increased prevalence with age. The main motor symptoms are resting tremor, bradykinesia, and rigidity, but they may be preceded by non-motor symptoms for over a decade [1]. The motor symptoms are usually unilateral at onset. There are no curative or disease-halting treatments to PD, but the quality of life of patients can be improved by relieving the symptoms with appropriate medication. Levodopa is the most efficient medication treatment for PD and it is normally used along with a combination of dopamine agonists and MAO-B inhibitors. Levodopa dosage must be increased as the disease progresses to maintain therapeutic response. Almost half of the patients develop motor fluctuations, dyskinesias, or both within five years after the initiation of levodopa treatment [2].

Deep brain stimulation (DBS) is currently the most efficient treatment for motor symptoms of advanced PD [3–5]. DBS treatment must be individually programmed for the best outcome. Programming of the DBS is an optimisation task that aims to minimise patients' symptoms while avoiding side effects that the stimulation may cause. The programming operator is usually a trained clinician. Even though there are strategies to ascertain the best DBS settings, the programming may be time consuming due to a multitude of parameters that require to be adjusted [5, 6]. Clinical evaluation during programming is based on the operator's observation and UPDRS-III score. While this is currently the best means to assess patients, subtask scores used to assess limb tremor and rigidity are coarse and there is variation in assessment among clinicians [7]. The symptoms vary on a daily basis and challenge the clinical assessment, which is typically performed within a short time frame. Thus, the operator has to rely partially on patients' own subjective assessment of the symptoms. The programming of DBS is becoming increasingly complicated due to the newest type of stimulators with customisable pulse types [8] and directional electrodes [9]. Objective measurement-based assessment of symptoms would be beneficial for optimising among different settings. Further, methods with the ability for long-term monitoring of the symptoms could be helpful aids in clinical decision-making.

Surface electromyographic (EMG) signals of PD patients have been found to have lower complexity and contain more rhythmic bursts and tonic background activity compared to signals of healthy controls [10–12]. Moreover, the muscle activation patterns of patients with PD also differ from those of patients with other motor diseases with similar symptoms [13–15]. Studies on elbow flexion-extension have reveal that EMG signals in PD contain more recurring patterns due to motor unit synchronization [10] and that DBS decreases these patterns and increases the sample entropy of EMG signals [16]. Although the effect of DBS on muscle activity has been studied in a few EMG-based studies [16, 17], the studies do not include DBS programming. Rissanen et al. report a difference in isometric EMG signals between lower and upper limbs during the programming of DBS [18]. EMG-based measurements accompanied with amplitude, frequency, and non-linear analysis have been used in the quantification of symptoms of PD [16, 19, 20].

EMG measurements are useful for evaluating muscle activity in patients with PD since they are non-invasive, easy to produce, and cost-effective. Thus, EMG measurement during dynamic arm movement could be a feasible method for assessing changes in muscle activity while programming DBS. In this study, the muscle activity of 13 patients with advanced PD were measured during the DBS programming session. The measurements were performed with seven different DBS settings that change the pulse voltage, frequency, and width within a clinical range. The aim of this work was to determine whether EMG measurement can quantify the difference among the DBS settings.

## Materials and methods

The study was approved by the Research Ethics Committee of the Northern Savo Hospital District. All patients provided written informed consent before the measurement. An EMG measurement was performed for 13 patients with advanced PD to observe the muscle activation patterns while adjusting the DBS. The demographics of the patients are presented in Table 1. The patients had implanted STN-DBS (Kinetra or Activa PC Neurostimulator, Medtronic Inc, Minneapolis, USA) because of severe motor fluctuations, wearing off phases, dyskinesias, tremor, or rigidity. At the time of measurement, the age of the patients was ($58 \pm 11$) (mean $\pm$ SD) and the duration from diagnosis ($11 \pm 5$) years. The severity of the motor

**Table 1. Patients, UPDRS-III score, and DBS details.** DBS months refer to number of months after DBS implantation. UPDRS-III assessment could not be completed on patients 2 and 13 due to side effects and notion $\geq$ is used.

| Patient | Age | Sex | UPDRS III on(off) | DBS months | | Optimal settings of DBS: voltage, frequency, pulse width |
|---|---|---|---|---|---|---|
| 1 | 46 | M | 21(36) | 5 | Right: | 3.5 V, 130 Hz, 60 μs |
| | | | | | Left: | 3.7 V, 130 Hz, 60 μs |
| 2 | 59 | F | 26($\geq$ 37) | 34 | Right: | 3.4 V, 130 Hz, 60 μs |
| | | | | | Left: | 3.2 V, 130 Hz, 60 μs |
| 3 | 64 | M | 22(29) | 23 | Right: | 3.1 V, 130 Hz, 60 μs |
| | | | | | Left: | 3.3 V, 130 Hz, 60 μs |
| 4 | 58 | F | 10(18) | 5 | Right: | 2.6 V, 130 Hz, 60 μs |
| | | | | | Left: | 2.5 V, 130 Hz, 60 μs |
| 5 | 64 | M | 16(36) | 2 | Right: | 2.8 V, 130 Hz, 60 μs |
| | | | | | Left: | 3.4 V, 130 Hz, 90 μs |
| 6 | 66 | M | 21(28) | 8 | Right: | 2.5 V, 130 Hz, 60 μs |
| | | | | | Left: | 2.5 V, 130 Hz, 60 μs |
| 7 | 66 | M | 34(45) | 21 | Right: | 2.3 V, 130 Hz, 60 μs |
| | | | | | Left: | 3.3 V, 130 Hz, 60 μs |
| 8 | 38 | M | 27(50) | 22 | Right: | 3.4 V, 130 Hz, 60 μs |
| | | | | | Left: | 3.4 V, 130 Hz, 60 μs |
| 9 | 71 | M | 22(36) | 4 | Right: | 3.1 V, 130 Hz, 60 μs |
| | | | | | Left: | 3.4 V, 130 Hz, 60 μs |
| 10 | 47 | M | 31(38) | 4 | Right: | 2.3 V, 180 Hz, 60 μs |
| | | | | | Left: | 2.5 V, 180 Hz, 60 μs |
| 11 | 58 | F | 12(23) | 6 | Right: | 2.4 V, 130 Hz, 60 μs |
| | | | | | Left: | 2.4 V, 130 Hz, 60 μs |
| 12 | 70 | M | 31(62) | 30 | Right: | 2.7 V, 130 Hz, 60 μs |
| | | | | | Left: | 3.3 V, 130 Hz, 60 μs |
| 13 | 45 | M | 31($\geq$ 36) | 29 | Right: | 3.1 V, 120 Hz, 60 μs |
| | | | | | Left: | 3.1 V, 120 Hz, 60 μs |

symptoms was assessed with UDPRS-III motor in the range of 0 – 108. Patients' UPDRS-III motor score with DBS off was (36 ± 12) and motor score with DBS on was (23 ± 8). The patients continued to have their current antiparkinsonian medication throughout the measurement. The measurements were performed at the BioMag laboratory, Helsinki, by an experienced neurologist (adjustment of DBS, assessment of UPDRS-III), and a physicist (EMG recordings).

## Measurement protocol

The surface of the biceps brachii muscle beneath the measurement electrodes was properly cleaned with wet ethanol cotton pads. Large disposable Ag/AgCl surface electrodes (Medicotest M-00-S) with an interelectrode distance of 3 cm were used to improve the signal quality and increase the number of recorded motor units. The electrodes were placed on top of left and right biceps brachii muscle, below the belly of the muscle. Bipolar configuration was used and the reference electrode was placed on an inactive point on the lateral side of the brachium, 6–7 cm (depending on the patient's arm size) from the recording electrodes. The signals were recorded with ME6000 biosignal monitor (Bittium Corporation, Oulu, Finland) with a sampling rate of 1000 Hz and resolution 1 µV.

The patients were made to sit on an ordinary wooden chair without armrests during the measurement. The task consisted of 7–8 repetitions of elbow flexion and extension, with elbow staying in place next to the torso. The movement began with the forearm parallel to the ground and had an angular range of approximately 80 degrees, depending on the patients mobility. The patient was guided in terms of the speed of the arm movement by showing them an example. A similar task has been used earlier to study EMG in PD [10, 21–23]. The movements of the left and right arms were measured separately. The patients were instructed on the course of the measurement beforehand. They were encouraged to get used to the measurement setup and practice the tasks before the measurement. If the patient felt unsure about the procedure, guidance was given during the measurement.

Before the study began, the patients' stimulators were programmed to optimal stimulation settings. The patients' original settings are presented in Table 1 and are subsequently referred to as the base setup. The patient measurement was always initiated with DBS at the base setup and ended with DBS off. After the first setup, the stimulation settings were changed one at a time and the measurement was repeated. After each adjustment of DBS, the patient's state was stabilised for a minimum of five minutes before beginning the measurement. The order of settings between the first and the last setting was randomised for each patient. The settings were changed relative to the patients' base setup, and the steps were similar to those used in an ordinary DBS programming session:

- base setup (A0)

- decrease pulse voltage by 0.3 V (-A)

- increase pulse voltage by 0.3 V (+A)

- decrease frequency by 30 Hz (-F)

- increase frequency by 30 Hz (+F)

- increase pulse width by 30 µs (+W)

- DBS off

The possible side effects were carefully observed and the patients were advised to immediately report subjective changes. If the patient experienced evident side effects or felt

uncomfortable, the base settings were immediately restored. The patients were given the choice to abort the measurement, but all patients were willing to pursue the measurements with other adjustments after the patients' clinical state was stabilised. The UPDRS-III motor score was determined in the beginning of the measurement with the base setup and with DBS off at the end. In other settings, only arm tremor and rigidity were assessed.

## Analysis

All the signal processing was performed in MATLAB 2019b (MathWorks Inc). Preprocessing of the EMG signal consisted of three stages of filtering. First, the smoothness priors detrending method [24] was used to remove low frequency variation from the signals. The method resembles a high-pass filter and the cut-off frequency of the filter is set by tuning a smoothing parameter—$\alpha$. Low frequency variation was eliminated by setting $\alpha = 300$, corresponding to a cut-off of approximately 10 Hz. Second, DBS-induced noise was eliminated with spectrum linear interpolation [25] of ± 2 Hz around the individual DBS stimulation frequency. Third, the harmonics of DBS stimulation frequency and possible other high frequency noise were eliminated with ninth-order Butterworth low-pass filter with a cut-off frequency of 150 Hz.

Each measurement consisted of seven to eight flexion extension repetitions, of which flexions were analysed in this study. The flexion phases of each repetition were manually selected from the signals. The parameters were calculated for each flexion patient-wise and then averaged.

The signal distribution and parameters based on fine structure were used to characterise the changes in EMG signals. The EMG signal of a healthy subject is a stochastic process in which single muscle activations are summed up to generate the desired type of muscle contraction. The motor symptoms of PD interfere with the EMG signal and introduce synchronisation of muscle activation, which is seen as tonic background and bursts in the signal [26]. The parameter KURT may be used to describe the the tonic background and the bursts of the signal, and can be defined for the EMG signal $x(t)$ as

$$KURT = E\left[ \left( \frac{x(t) - \mu}{\sigma} \right)^4 \right], \tag{1}$$

where $\mu$ is the mean and $\sigma$ the standard deviation of $x(t)$. EMG signal kurtosis is typically higher in patients with PD compared to healthy subjects due to the higher number of signal peaks.

The fine structure of the signal can be analysed with recurrence rate (%REC) and correlation dimension (CD). These parameters originate from recurrence quantification analysis. %REC describes the rate of recurrent structures in the signal and can be described in the following manner:

$$\%REC = \frac{1}{N^2} \sum_{i,j=1}^{N} \mathbf{R}(i,j) = \frac{1}{N^2} \sum_{i,j=1}^{N} \Theta(\epsilon - d_{ij}) \tag{2}$$

where $N$ is the length of the signal, $\Theta$ the Heaviside step function, $\epsilon$ the calculation threshold 0.2, $d_{ij}$ the Euclidean distance between two signal elements and $\Theta(\epsilon - d_{ij})$ describes the recurrent element. Webber et al. described %REC for the assessment of dynamic systems in 1994 [27]. Recurring patterns in EMG signals arise due to increased synchronisation of muscle activation potentials. A larger number of recurrent EMG patterns have been observed during isometric muscle contraction in patients with PD compared to healthy subjects [26]. Moreover, the %REC has also been used to analyse dynamic muscle contractions in PD [10].

The correlation dimension (CD) can be defined as boundary value from similar equation when autocorrelation $i = j$ is ignored and $\epsilon$ and signal length $N$ approach infinity:

$$\mathrm{CD} = \lim_{\epsilon \to \infty} \lim_{N_m \to \infty} \log \left( \frac{1}{N^2} \sum_{i \neq j = 1}^{N} \Theta(\epsilon - d_{ij}) \right) \frac{1}{\log \epsilon}. \tag{3}$$

The CD cannot be determined without infinite signal, but it can be estimated by determining the slope of the logarithmic regression line. The EMG correlation dimension has been previously linked to muscle fatigue [28]. The correlation dimension measures the complexity of the signal and has been observed to be lower in the isometric EMG of patients with PD compared to healthy controls [26]. A decrease in correlation dimension indicates that the complexity of the signal decreases. In PD, this is typically caused by periodicity due to synchronisation.

The differences in EMG parameters between the different DBS settings were individually compared against the base setup. The patients' more symptomatic arm was determined based on the anamnesis and the UPDRS-III score. Lilliefors test for data normality was performed for the parameters. The parameters were not normally distributed and Wilcoxon signed rank test was performed to determine if the differences between setups compared to the base setup were significant.

## Results

The UPDRS-III score decreased from 15% to 45% (mean 35%) when comparing DBS off and the base setup (Table 1). Measured arm tremor and rigidity scores are presented in Table 2. Arm tremor or rigidity did not change significantly between the settings. In three patients, no changes in arm tremor or rigidity occurred during the entire measurement.

Fig 1 presents the EMG signal response of a patient during the adjustment of DBS. The response to DBS stimulation varied with each individual. On this patient the activation patterns during the contraction of biceps muscle are regular when DBS is set to the base setup (A0). The change of stimulation voltage caused burst patterns (+A) and a large spike (-A) probably caused by a dystonic movement at the end of the extension of the elbow. Changing the pulse frequency prolonged the flexion part of the signal. When the stimulator was turned off, the EMG activity increased notably and did not cease between the contractions. Instead, a tonic background muscle contraction remained throughout the task with more synchronised bursts in the signal.

Significant changes were observed during the adjustment of DBS in the parameters calculated from the patients' more symptomatic arm (Table 3). %REC increased significantly in all

**Table 2. Arm tremor and rigidity during settings (mean ± sd).** The changes in arm tremor and rigidity were non-significant throughout the measurement. The arm tremor and rigidity increased due to a decrease in stimulation values (-A, -F, off) and decreased due to an increase in stimulation values (+A, +F) in certain patients. The changes were generally small compared to the deviation.

| Setting | Tremor (0–8) | Rigidity (0–8) |
|---------|--------------|----------------|
| A0 | 0.4 ± 0.8 | 1.2 ± 1.5 |
| -A | 1.0 ± 1.5 | 1.4 ± 1.9 |
| +A | 0.1 ± 0.3 | 0.4 ± 0.9 |
| -F | 1.1 ± 1.4 | 1.1 ± 1.4 |
| +F | 0.2 ± 0.4 | 0.8 ± 1.4 |
| +W | 0.4 ± 1.0 | 0.3 ± 0.7 |
| DBS off | 1.6 ± 2.0 | 2.2 ± 2.0 |

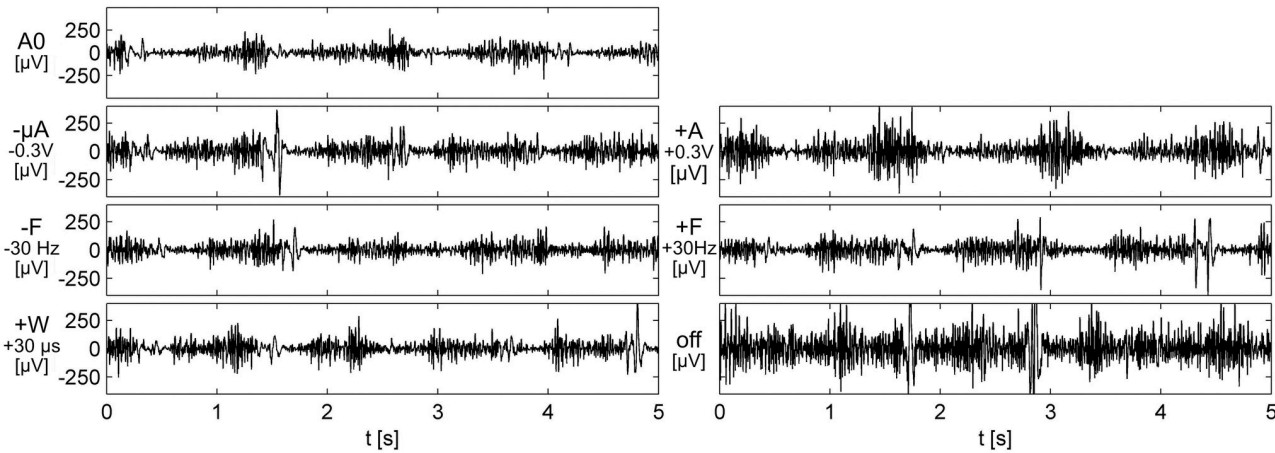

**Fig 1. The EMG signals of one patient during the elbow flexion-extension task during the adjustment of DBS.** The morphology of the EMG signal differs among the different adjustments of DBS. Turning the stimulator off caused the strongest effect and introduces significant tonic background to the EMG signal. Scaling is identical in all settings.

settings compared to the base setup. Similar changes were observed for KURT and CD, but the changes were significant in only part of the setups. KURT increased significantly in +A, -F, and DBS OFF, while CD decreased significantly in +A, -F, +F, and +W. The changes in the parameters calculated from the patients' less symptomatic arm were non-significant during the adjustment of DBS. %REC and CD differed the most when the pulse width was adjusted (+W), despite the fact that a major proportion of the patients could not be measured due to strong effects of DBS. There was also a considerable amount of variation between the patients in the parameters in +W as well as in certain parameters in -A, -F, and +F (Fig 2). An increase or decrease in pulse frequency affected EMG parameters slightly more than an increase or decrease in pulse voltage.

Side effects were observed 17 times, of which 11 caused the abortion of the measurement as it caused the patient discomfort. Most of the side effects were caused by increasing the pulse

**Table 3. Relative EMG signal parameters recurrence rate (%REC), kurtosis (KURT), and correlation dimension (CD) (median ± IQR).** %REC changed significantly after each adjustment. Increasing voltage or decreasing frequency of the stimulation changed each parameter significantly. Significance levels $^*p < 0.05$, $^†p < 0.01$.

| Setting | %REC | KURT | CD |
|---|---|---|---|
| *More affected hand* | | | |
| -A (-0.3 V) | $1.10 \pm 0.50^*$ | $1.02 \pm 0.16$ | $0.96 \pm 0.11$ |
| +A (+0.3 V) | $1.34 \pm 1.14^*$ | $1.06 \pm 0.19^*$ | $0.91 \pm 0.22^*$ |
| -F (-30 Hz) | $1.65 \pm 0.43^†$ | $1.13 \pm 0.23^*$ | $0.88 \pm 0.11^†$ |
| +F(+30 Hz) | $1.66 \pm 0.79^*$ | $1.13 \pm 0.25$ | $0.92 \pm 0.09^*$ |
| +W (+30 μs) | $1.98 \pm 1.69^*$ | $1.13 \pm 0.23$ | $0.82 \pm 0.22^*$ |
| DBS off | $1.47 \pm 1.10^*$ | $1.10 \pm 0.08^†$ | $0.94 \pm 0.15$ |
| *Less affected hand* | | | |
| -A (-0,3 V) | $0.90 \pm 0.49$ | $0.99 \pm 0.10$ | $1.02 \pm 0.11$ |
| +A (+0,3 V) | $1.02 \pm 0.73$ | $1.01 \pm 0.14$ | $0.95 \pm 0.15$ |
| -F(-30 Hz) | $1.06 \pm 0.45$ | $0.97 \pm 0.22$ | $0.96 \pm 0.14$ |
| +F(+30 Hz) | $0.92 \pm 0.70$ | $0.96 \pm 0.22$ | $1.01 \pm 0.17$ |
| +W (+30 μs) | $0.95 \pm 0.68$ | $1.01 \pm 0.27$ | $0.96 \pm 0.14$ |
| DBS off | $1.32 \pm 0.66$ | $1.01 \pm 0.25$ | $0.95 \pm 0.14$ |

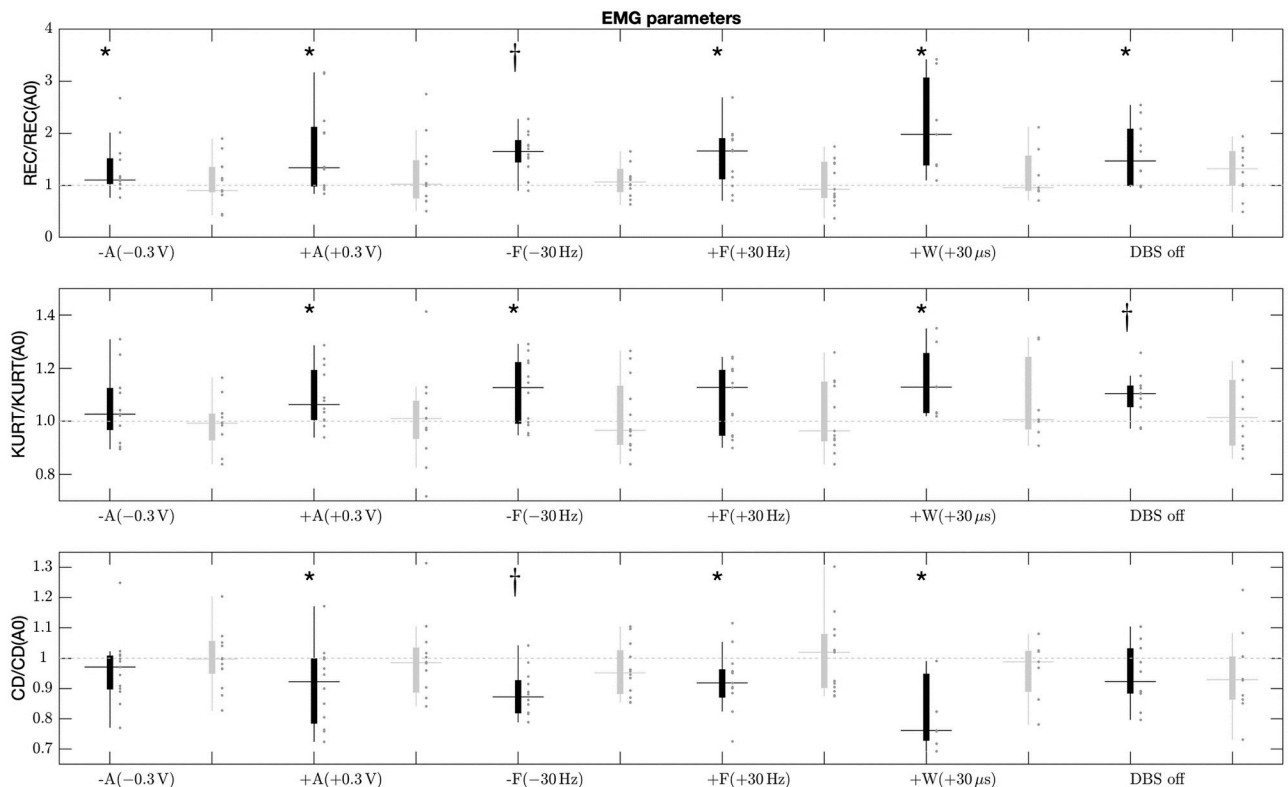

**Fig 2. The EMG parameters of patients during elbow flexion-extension task during different adjustments of DBS.** There are significant changes in each parameter compared to the base setup. Recurrence rate changes significantly in each adjustment. Changes are stronger on more symptomatic side compared to less symptomatic side. Fig 2 shows parameter vaules separately for more (black) and less (gray) symptomatic arm. The small gray dots next to boxplot indicate the individual patient values. The vertical axis value 1 is the setting A0 value. The deviation is considerable in all settings. Significance levels $^*p < 0.05$, $^\dagger p < 0.01$.

width. Six patients developed dysarthria probably by unwanted stimulation of corticobulbar fibers. Three patients had muscle contraction probably due to stimulation of corticospinal fibers. One patient developed diplopia due to stimulation of oculomotor nerve. Seven patients developed dyskinesia due to non-optimal stimulation. All side effects vanished when original DBS settings were restored.

## Discussion

Muscle activation patterns of 13 patients with advanced PD were measured during adjustment of DBS. The DBS was adjusted in small steps of 0.3V, 30Hz, and 30μs, similar to a clinical DBS programming session. The main finding was that adjustment of the DBS causes changes to the EMG signal morphology via changed muscle activation patterns. The respective clinical markers for rigidity or tremor did not change significantly. The secondary finding was that changes in signal morphology were more pronounced on the patients' more symptomatic arm, whereas there were no significant changes in signal morphology on the patients' less symptomatic arm.

The EMG parameters (%REC, KURT, CD) differed significantly between the optimal and other adjustments of DBS and indicated that with optimal settings, the signal contained less parkinsonian signal features on average compared to any other measured setup. Adjusting DBS voltage, frequency, or pulse width into other than the optimal values, the EMG activity changed and contained more synchronised bursts on average, which are known to be related

to the symptoms of PD. In this study, %REC was the most sensitive parameter for detecting differences between the base setup and the other DBS adjustments. %REC measures the density of repeating patterns in EMG signal and is typically higher in patients with PD compared to healthy subjects [26]. KURT and CD differed between the adjustments, but the difference was not significant in each setup. These parameters describe the signal in a different manner compared to %REC and it appears that recurrent patterns are more sensitive indicators for DBS adjustments compared to signal complexity and peakedness.

Montgomery proposes that response to DBS stimulation follows a U-shaped curve: increase in stimulation voltage improves motor performance only until a certain point is reached. After this point, according to Montgomery, patient's motor performance becomes worse if the voltage is further increased [29]. The results of the study are in agreement with Montgomery's theory: the EMG parameters had their extremum value at the base setup (low peak for %REC and KURT, high peak for CD).

Although the parameters changed significantly at the group level, there was considerable variation. This was taken into account by normalising the parameters with the patients base setup, thereby allowing for a comparison of the changes among the patients. At the individual level, the parameters differed between the optimal and non-optimal stimulation in most patients. In certain patients, the parameters did not reach the peak value at base setup and, thus, the parameters did not indicate the optimal stimulation setting. This was observed slightly more when stimulation amplitude was altered compared to frequency or pulse width adjustment. It must be noted that altering the stimulation settings may improve some PD symptoms like rigidity, but simultaneously induce unwanted motor or non-motor side effects.

Rigidity and tremor are suggested for assessing DBS symptomatic relief during programming as they react quickly to changes in adjustment [6]. Rigidity is more reliable symptom for adjustment compared to tremor since tremor often fluctuates unlike rigidity. While the UPDRS-III full motor assessment differed significantly between the base setup and DBS off, there was no significant difference in arm tremor and rigidity tasks between the DBS settings. This may be due to the following factors: 1) DBS may affect symptoms other than arm tremor and rigidity, 2) the scoring for arm tremor and rigidity may be too coarse to detect minor changes due to adjustment of DBS in small steps. Similar results have been reported by Heida et al. [30]. The UPDRS evaluation has been devised for comprehensive assessment of motor and non-motor symptoms of the PD, while single tasks score were used to assess patients in this study. Even though there was no difference in rigidity and tremor score between the adjustments, the patients still preferred the base settings over other adjustments. The optimal settings for DBS depend on multiple factors, including the possible side effects that DBS may cause. Full UPDRS-III evaluation was not performed with each DBS setting, as it would have been too time-consuming and too strenuous for patients.

The finding that EMG parameters change significantly during adjustment of DBS while arm rigidity and tremor do not is significant. The result is in concordance with Heldman et al. [7], who suggest that kinematic measurement can be more sensitive for characterising finger tapping in PD compared to assessment by a clinician. The changes in the EMG parameters reflect the small steps in which stimulator was adjusted. If larger adjustment steps were used, the differences would possibly have been more pronounced. Despite adjusting the stimulator in small steps, motor and non-motor side effects were observed. Dyskinesia, dysarthria, or impairment of vision are typically caused by unwanted stimulation of nearby tissue. Rapid movements of limbs require strong muscle activation and cause significant amplitude spikes to the EMG signal. Moreover, tonic muscle contractions may be seen as background activity in EMG even at rest. Motor side effects were rarely recorded during the study since the measurement was aborted after their emergence. The adjustment of the pulse width caused the greatest

number of side effects and thus, the patient in Fig 1 is not an exemplary case, since EMG morphology changed only slightly after pulse width adjustment. Other generalisations regarding what side effects were caused by different stimulation settings cannot be made based on our results.

The therapeutic effect of DBS has different delay for relieving different symptoms. Rigidity and tremor are relieved in seconds to minutes [6, 17, 31], while it may take from minutes to days to relieve bradykinesia and axial symptoms [6]. This is an inherent challenge in DBS programming and it is possible that symptoms with longer stabilisation time cannot be observed during the programming session to the full extent. These symptoms must be observed in multiple sessions or with long period measurements. A stabilisation time of minimum 5 minutes was selected to maintain the measurement session short—a total of 2,5 hours. While it is possible that the patients' response to the DBS was still stabilising, most of the effects were likely present during the measurement.

The patients had advanced PD with motor symptoms that could not be adequately controlled by optimal medical treatment. Antiparkinsonian medication is typically used along with DBS therapy to achieve optimal control of symptoms. While the first adjustments of the stimulator after installation are done without medication, the fine-tuning may be done with medication. The patients were studied with their current normal medication. While this might weaken the results, a study suggests that medication cannot fully alleviate patterns typical to PD from EMG signals [21]. The effect of medication was taken into account in the planning by keeping the total duration of the measurements as low as possible, while maintaining a sufficiently long time for DBS stabilisation. This helped in two ways: the measurements were not burdensome to the patients, but also the medication response was relatively stable during the measurement. Further, the different DBS settings were measured in randomised order to decrease systematic errors caused by a change in the medication response.

## Conclusion

Clinical observation is currently the golden standard for PD diagnosis and is predominantly used for programming the DBS. This study is one of the first studies to evaluate EMG signal parameters during the adjustment of DBS. The results suggest that while UPDRS-III tremor and rigidity tasks are used a part of the evaluation of programming the DBS, these scores may not be sufficiently specific to detect the small differences in the patients motor state and thus the operators, despite being trained professionals, may have to rely on their craftmanship for the evaluation of tremor and rigidity, since the UPDRS motor scoring may lack precision.

This emphasises the need for specific objective methods to assess the symptoms of PD during DBS programming. While this study shows promising results for using EMG to quantify changes during the adjustment of DBS, the role of the study was a proof of concept and shall be validated with a larger number of patients. A larger number of patients would enable, for example, dividing the patients into groups based on their main symptom (tremor, rigidity, motor fluctuations), examining only the patients with a change in the clinical tremor or rigidity, and performing receiver operator characteristic analysis or other sophisticated statistical predictions.

## Supporting information

**S1 Data.**
(XLSX)

## Author Contributions

**Conceptualization:** Verneri Ruonala, Eero Pekkonen, Olavi Airaksinen, Markku Kankaanpää, Pasi A. Karjalainen, Saara M. Rissanen.

**Data curation:** Verneri Ruonala.

**Formal analysis:** Verneri Ruonala, Pasi A. Karjalainen.

**Funding acquisition:** Olavi Airaksinen, Markku Kankaanpää, Pasi A. Karjalainen.

**Investigation:** Verneri Ruonala, Eero Pekkonen.

**Methodology:** Verneri Ruonala, Eero Pekkonen, Olavi Airaksinen, Markku Kankaanpää, Saara M. Rissanen.

**Project administration:** Olavi Airaksinen, Markku Kankaanpää, Pasi A. Karjalainen, Saara M. Rissanen.

**Resources:** Eero Pekkonen, Saara M. Rissanen.

**Supervision:** Pasi A. Karjalainen, Saara M. Rissanen.

**Validation:** Olavi Airaksinen, Saara M. Rissanen.

**Visualization:** Verneri Ruonala.

**Writing – original draft:** Verneri Ruonala.

**Writing – review & editing:** Verneri Ruonala, Eero Pekkonen, Olavi Airaksinen, Markku Kankaanpää, Pasi A. Karjalainen, Saara M. Rissanen.

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
