## [Decision Letter · Decision Letter 0]

23 Jun 2021

PONE-D-21-16366

Elbow flexion EMG morphology changes during adjustment of deep brain stimulator in advanced Parkinson's disease

PLOS ONE

Dear Dr. Ruonala,

Thank you for submitting your manuscript to PLOS ONE. After careful consideration, we feel that it has merit but does not fully meet PLOS ONE’s publication criteria as it currently stands. Therefore, we invite you to submit a revised version of the manuscript that addresses the points raised during the review process.

We look forward to receiving your revised manuscript.

Kind regards,

Karsten Witt

Academic Editor

PLOS ONE

Journal Requirements:

3. We note that you have a patent relating to material pertinent to this article. Please provide an amended statement of Competing Interests to declare this patent (with details including name and number), along with any other relevant declarations relating to employment, consultancy, patents, products in development or modified products etc. Please confirm that this does not alter your adherence to all PLOS ONE policies on sharing data and materials, as detailed online in our guide for authors http://journals.plos.org/plosone/s/competing-interests by including the following statement: "This does not alter our adherence to PLOS ONE policies on sharing data and materials.” If there are restrictions on sharing of data and/or materials, please state these. Please note that we cannot proceed with consideration of your article until this information has been declared.

Reviewers' comments:

Reviewer's Responses to Questions

**Comments to the Author**

1. Is the manuscript technically sound, and do the data support the conclusions?

Reviewer #1: Yes

Reviewer #2: Partly

2. Has the statistical analysis been performed appropriately and rigorously? 

Reviewer #1: No

Reviewer #2: I Don't Know

3. Have the authors made all data underlying the findings in their manuscript fully available?

Reviewer #1: No

Reviewer #2: No

4. Is the manuscript presented in an intelligible fashion and written in standard English?

Reviewer #1: No

Reviewer #2: No

5. Review Comments to the Author

Reviewer #1: General comments

This manuscript investigates the effect of modifying DBS settings on EMG of the biceps brachii muscle bilaterally and the UPDRS III clinical sub-scores for tremor and rigidity. Results indicate that whilst clinical sub-scores did not change between settings, there were significant differences in EMG metrics. This is an interesting and relevant study to better understand if and how motor changes relate to modifying DBS frequency, width, and amplitude settings. There are however several general areas that need to be addressed:

• There are numerous grammatical errors throughout the manuscript usually with a word missing, incorrectly used word and several typos. Please proofread the manuscript and correct.

• No discussion of how altered ‘EMG signal morphology’ relates to clinical changes has been made. Is there any evidence that for example increased recurrence rate (greater repetition) relates to improved clinical function?

• Complexity has a definite mathematical definition and indicates greater interactions (Didier Delignieres & Vivien Marmelat, Fractal Fluctuations and Complexity: Current Debates and Future Challenges, December 2012, Critical Reviews in Biomedical Engineering 40(6):485-500). Decreased regularity does not mean increased complexity. Please check throughout the manuscript that the use of the word ‘complexity’ is appropriate.

• Tremor and rigidity were assessed with sub-scores of UPDRS III which is a very blunt tool. Tremor can be measured quantitatively with an accelerometer and an indication of rigidity by measuring co-activation of agonist and antagonist. Why were quantitative measures of recording tremor and rigidity not undertaken?

• Statistics is only very briefly mentioned. Only a Wilcoxon signed rank test was performed. Were the data nonparametric? Could Friedman/ Kruskal Wallis be used? Please expand more on the statistics.

Methods

• In Methods, how was the location of placement of the electrodes determined? Were SENIAM guidelines followed?

• At what angle was the elbow joint at the beginning of the recording session and what was the angular range of movement and velocity? Length of fascicle, angular velocity and eccentric/ concentric contractions will all modify EMG signals.

• Please include the safety levels for the DBS parameters.

• Please provide a reference on pg4, line 124 for why 5 minutes was selected between setings.

Analysis

• How were the flexion phases of the EMG signals selected?

• Please explain why kurtosis, recurrence rate and correlation dimension were analysed and not other metrics. What algorithms were used and what software?

• Results

• A demographics’ table with individual’s demographics and clinical details should be included

Discussion

• Line 217 – Is it possible for anything to change ‘instantly’? Please revise.

• ‘Miniscule’ is a vague term and preferable to be replaced with ‘nonsignificant changes’.

• Lines 238-240. EMG records muscle activity therefore by definition it will have greater sensitivity to muscle activity changes as this is what it is measuring! Please revise. EMG is quantitative and UPDRS III is a subjective qualitative clinical measure.

• Line 240 – How is ‘optimally tuned’ DBS defined?

Conclusion

• The conclusion is weak as there is no strong message other than changes occur. It is recommended to introduce some clinical relevance.

Figures

Figure 1- Please either define the acronyms of the 7 settings in the figure legend or ideally, add as subheadings.

Figure 2 – See above.

Reviewer #2: The authors address one of the main challenges in deep brain stimulation (DBS) therapy: To determine DBS parameters leading to best clinical efficacy of DBS in the individual patient by using an objective feedback measure. Such an approach may not only spare time needed during “conventional” DBS programming sessions but may also reduce the number of recurrent clinical visits normally needed to evaluate often delayed DBS effects. Moreover, individual DBS clinical efficacy may be enhanced and DBS side effects reduced.

The authors present an explorative study which evaluates measures of EMG recordings during a patient driven phasic motor task of an elbow flexion correlated to clinical effects of subthalamic nucleus DBS in Parkinson´s disease (PD) patients.

Some additional aspects may strengthen the findings and could help to improve the manuscript:

Introduction

- The introduction should be shortened and should have stronger focus on the need to improve the quality of programming algorithms for DBS therapy.

- Description of general medical treatment strategies in PD may be left out as well as hypotheses on the mechanisms of action of DBS.

- Instead, a statement that DBS is an appropriate therapeutic option in late stage PD may be enough and may be followed by discussing the problems clinicians and patients are confronted with during conventional DBS programming sessions (try and error, delayed DBS effects and side effects).

- The last paragraph in the introduction may then illustrate why the authors have chosen EMG recordings to objectify clinical DBS effects.

Methods

- Authors should explain why they have chosen the mentioned motor task. To me, this task is far from being “objective” as the performance of such a task is patient driven in acceleration and speed of the movements and in its muscle strength which directly affects EMG activity.

- The authors should explain why they have chosen to evaluate such minimal changes of amplitude and frequency compared to the clinically chosen DBS settings, which the authors call “base setup”. I do agree that sometimes subtle changes of DBS parameters may influence motor symptoms of PD but I cannot see any rationale for +-0.3V, +-30Hz, +-30µs.

- I would strongly recommend to additionally study patients in medication / dopamine depleted state as PD motor symptoms may similarly become reduced with medication or DBS. Otherwise, their relative influence on symptom relief cannot be described and therefore effects on EMG recording cannot clearly be assigned to one or the other therapy: What parameters of EMG recordings are influenced by medication? What parameters of EMG recordings are influenced by DBS?

- Characteristics of EMG recordings due to side effects (affecting pyramidal tract) should be defined to distinguish “optimal” from “above threshold” stimulation. Especially because the authors describe “side effects” (without further specification of their clinical appearance) due to greater pulse widths.

Analysis

- The reader may profit from a more detailed clinical / practical view on and explanation of the parameters chosen to be evaluated from EMG recordings: kurtosis, recurrence rate, correlation dimension.

- What are the clinical significances of differences in kurtosis, recurrence rate and correlative distribution between PD and healthy controls? What is the physiological meaning of the mentioned parameters?

Results

- Table 1 shows that “base setup” may not be “best setup” as the increase of the parameters amplitude, frequency or pulse width, even in such narrow margins (see above), may further improve DBS clinical efficacy. Authors should discuss and may further analyse differences in EMG recordings which may distinguish patients with suboptimal stimulation from optimal stimulation (e.g. complete clinical rigidity control as clinical feeback).

- Figure 2 shows results of EMG-parameters in relation to “base setup” (A0). Although clinically more effective to weaken tremor and rigidity (see table 1), A+ is still greater 1, suggesting A+ to be “worse” than A0. The authors may explain 1) why they have chosen to analyse EMG parameters relative to A0 and 2) why A+ performs less effective than A0 in the EMG parameters although clinically better in the reduction of PD motor symptoms (same for F+ which performs worse than DBS OFF in the EMG parameters, although of better clinical efficacy than A0).

- The authors should explain and name the mentioned “side effects” of DBS due to changed parameters (e.g. pulse width). DBS may have affected the fibres of the pyramidal tract? What are the effects of these “side effects” on EMG recordings? Again, authors may analyse / discuss how to distinguish suboptimal, optimal and above threshold stimulation (side effects) by means of EMG recordings.

- Statistical results are not rigorously stated (performed tests and resulting values are missing).

Discussion

- The “U-shaped theory” of optimal stimulation parameters needs to be better explained. “EMG parameters had their extremum at the base setup” is not shown in the results section (only relative values in fig2).

The whole manuscript needs major editing concerning language and spelling.

6. PLOS authors have the option to publish the peer review history of their article (what does this mean?). If published, this will include your full peer review and any attached files.

Reviewer #1: **Yes: **Annette Pantall

Reviewer #2: No

---

## [Author Response · Author response to Decision Letter 0]

3 Oct 2021

Dear Editor, reviewers #1 and #2,

Thank you for your excellent corrections and comments. We have revised the manuscript thoroughly.

Dear Editor,

>>>Please ensure that your manuscript meets PLOS ONE's style requirements, including those for file naming. The PLOS ONE style templates can be found at

...PLOSone_formatting_sample_main_body.pdf and

...PLOSone_sample_title_authors_affiliations.pdf

 Manuscript has been checked against to these templates and required changes have been made. However, requirement "Tables must be cell-based in Microsoft Word or embedded with Microsoft Excel" could not be fullfilled as the manuscript is prepared on PLOS ONE LaTeX template.

>>> We note that the grant information you provided in the 'Funding information' and 'Financial Disclosure' sections do not match.

 Funding information updated:

 The work was supported by Academy of Finland under project (252748). VR has received research grant from Finnish Parkinson foundation. EP has received Finnish Govermental research funding (TYH-fund). The funders had no role in study design, data collection and analysis, decision to publish, or preparation of the manuscript.

#>>> We note that you have a patent relating to material pertinent to this article. Please provide an amended statement of Competing Interests to declare this patent (with details including name and number), along with any other relevant declarations relating to employment, consultancy, patents, products in development or modified products etc.

 V.R. is an inventor in patent application PCT/FI2019/050163 "Measurement unit and monitoring system for monitoring indicator of Parkinson's disease in person".

 S.M.R. and P.A.K. are inventors in patent applications EP18159445.8 "Electrode patch, system, and method for detecting indicator of Parkinson's disease in person", PCT/EP2019/055002 "Electrode patch, system, and method for detecting indicator of Parkinson's disease in person", and PCT/FI2019/050163 "Measurement unit and monitoring system for monitoring indicator of Parkinson's disease in person".

 S.M.R and P.A.K are co-founders of Adamant Health Ltd.

 E.P. is a Standing Member of the MDS Non-Motor Parkinson's Disease Study Group.

>>> Please confirm that this does not alter your adherence to all PLOS ONE policies on sharing data and materials, as detailed online in our guide for authors by including the following statement... .

 These patents, patent applications nor other interests do not alter our adherence to PLOS ONE policies on sharing data and materials.

If there are restrictions on sharing of data and/or materials, please state these. Please note that we cannot proceed with consideration of your article until this information has been declared.

 This study is part of larger research project that has been approved 2004 and updated 2011 by research ethics committee of the Northern Savo Hospital District (155/2004). The approval of the study requires that all the raw patient data related to the research project has to be destroyed after the end of the project. The ethics statement related to this project allows us to publish processed data, but the data has to be anonymised so that it cannot be linked to the patients participating the study. The data regulations in Finland are strict and all health related data of patients is considered confidential by definition. Please note that Finland is not that big a country, and there are not so many patients with Parkinson's disease, we have to be especially considerate on this matter. 

 We are unfortunately unable to share raw data from the measurements. However, we do want to cooperate to solve this issue within the limits set by the ethics statement, national law and the EU directive on data protection (GDPR). We have included a datasheet with intermediate data to verify the results and conclusions stated in the manuscript. We do hope that this helps the decision.

Dear reviewer #1,

>>> There are numerous grammatical errors throughout the manuscript usually with a word missing, incorrectly used word and several typos. Please proofread manuscript and correct.

 Manuscript updated:

>>> No discussion of how altered ‘EMG signal morphology’ relates to clinical changes has been made. Is there any evidence that for example increased recurrence rate (greater repetition) relates to improved clinical function?

 We like to think it other way around - improved clinical function is reflected to muscle activation as change in parameters e.g. recurrence. The parameters in this study have been shown to differ between patients with PD and healthy subjects (Meigal et al. 2009, Rissanen et al. 2008).

 As you have pointed out later in a comment, UPDRS subtask scores are a blunt tool for assessing delicate changes. As the results of the study show, this clinical marker is unable to differentiate between the settings even though the patients are suffering from advanced PD. Thus, the adjustment has to be based on other factors as well. During the course of these measurements, different stimulation settings were tested, but as a whole, patients had least symptoms and side effects with the optimal setup: after all the patients chose to continue with the same optimal settings after the adjustment.

 Manuscript updated.

>>> Complexity has a definite mathematical definition and indicates greater interactions (Didier Delignieres & Vivien Marmelat, Fractal Fluctuations and Complexity: Current Debates and Future Challenges, December 2012, Critical Reviews in Biomedical Engineering 40(6):485-500). Decreased regularity does not mean increased complexity. Please check throughout the manuscript that the use of the word ‘complexity’ is appropriate.

 Manuscript updated. Decreased complexity now refers only to decrease in correlation dimension, which is a measure of complexity.

>>> Tremor and rigidity were assessed with sub-scores of UPDRS III which is a very blunt tool. Tremor can be measured quantitatively with an accelerometer and an indication of rigidity by measuring co-activation of agonist and antagonist. Why were quantitative measures of recording tremor and rigidity not undertaken?

 Excellent point, this is one of the key messages of this study. UPDRS-III subtasks, despite being blunt, are still the golden standard when assessing PD symptoms during adjustment of DBS. The scope of this study was to prove that the changes that DBS causes to muscle activation can be detected objectively with simple EMG measurement setup. We are aware, that tremor can be measured with accelerometer and it has been proven in many peer reviewed publications. The measurement of rigidity, however, we believe is not that straightforward and depends on many factors that may include agonist-antagonist activation. Biceps EMG has been used to recognize rigidity in earlier study with similar task (Rissanen et al. 2009).

 Manuscript updated.

>>> Statistics is only very briefly mentioned. Only a Wilcoxon signed rank test was performed. Were the data nonparametric? Could Friedman/ Kruskal Wallis be used? Please expand more on the statistics.

 Lilliefors test for data normality was performed for the parameters. Wilcoxon test was used because it is paired and determines directly the significance between the measurement phases. Kruskal-Wallis is used for non-paired data and while it may be used, part of the information is lost in the process.

 Manuscript updated.

Methods

>>> In Methods, how was the location of placement of the electrodes determined? Were SENIAM guidelines followed?

 SENIAM guidelines were followed in positioning recording electrodes. SENIAM guideline on reference electrode positioning was not followed. The analysis methods that are used in this study require high quality EMG measurements. To improve signal quality and reduce noise, we chose to use a EMG preamplifier with approx 15 cm electrode leads. With this setup, lateral side of brachium was an optimal place to position the reference electrode. SENIAM guideline on electrode separation was not followed. There are theoretical and practical reasons to this. Electrode size and the distance between them affect to the number of motor units they record. Since this study is about EMG morphology and motor unit synchronisation patterns, it was necessary to measure multiple motor units simultaneously. Due to this big (diameter 3 cm) electrodes with 3 cm separation were used. The electrodes could have been cut to decrease the distance, but that would have risked to shortcut the electrode gels and also would have decreased the number of motor units they measure. The measurement setup has been used in numerous studies (Rissanen et al. 2007, 2009 and 2011, Meigal et al. 2009 and 2012, Ruonala et al. 2014 and 2018).

 Manuscript updated.

>>> At what angle was the elbow joint at the beginning of the recording session and what was the angular range of movement and velocity? Length of fascicle, angular velocity and eccentric/ concentric contractions will all modify EMG signals.

 The elbow angle was not measured during the measurements. The measurement began with forearm parallel to ground. Angular range of the movement was approx 80 degrees and the repetition frequency 45-55 bpm.

 Manuscript updated.

>>> Please include the safety levels for the DBS parameters.

 Usually DBS voltage range up to max 5 volts is recommended. The stimulation pulse width and stimulation frequency depend on type of stimulator and the stimulation target. For STN-DBS 30-210 µs have been studied. Stimulation frequency over 185 Hz does not improve symptoms significantly. (Koeglsperger et al 2019)

 During DBS programming in clinic, we at first keep pulse width and frequency constant at 60 µs and 130 Hz, respectively. First monopolar survey will be accomplished by gradually increasing voltage by 0.5 V steps up to 5 V, continuously observing clinical response (rigidity being most reliable followed by bradykinesia and tremor) unless clear side-effects will appear, like dysarthria, diplopia, paresthesia. When necessary, specifically in tremor dominant PD patients’ response to frequency will be tested between 60-200 Hz. Usually 60, 130, 160 and 200 Hz. Finally response to pulse width will be tested by gradually increasing pulse width up to 90 µs (STN). Activa/Kinetra system do not allow to reduce pulse width below 60 µs.

 During this study, one patient was tested for 210 Hz stimulation frequency, one patient was tested for 120 us pulse width. Neither of these settings did not produce better results compared to optimal settings.

 Manuscript updated.

>>> Please provide a reference on pg4, line 124 for why 5 minutes was selected between setings.

 Manuscript updated: Discussion on stabilisation time has been transferred to 'Discussion' section. Detailed explanation with references is included.

Analysis

>>> How were the flexion phases of the EMG signals selected?

 Flexion phases of the movement were selected by hand from the signals by the author.

>>> Please explain why kurtosis, recurrence rate and correlation dimension were analysed and not other metrics. What algorithms were used and what software?

 Kurtosis, recurrence rate and correlation dimension showed strongest potential in detecting DBS induced changes. The parameters were caluculated by algorithms that were tailored for this study and are based on the equations in 'Analysis' section. The computations were made with MATLAB 2019b (Mathworks).

 Manuscript updated: Computation software added.

Results

>>> A demographics’ table with individual’s demographics and clinical details should be included

 Manuscript updated.

Discussion

>>> Line 217 – Is it possible for anything to change ‘instantly’? Please revise.

 Manuscript updated.

>>> ‘Miniscule’ is a vague term and preferable to be replaced with ‘nonsignificant changes’.

 Manuscript updated.

>>> Lines 238-240. EMG records muscle activity therefore by definition it will have greater sensitivity to muscle activity changes as this is what it is measuring! Please revise. EMG is quantitative and UPDRS III is a subjective qualitative clinical measure.

 Manuscript updated.

>>> Line 240 – How is ‘optimally tuned’ DBS defined?

 Optimally tuned refers to base setup defined in 'Methods' section.

 Manuscript updated.

Conclusion

>>> The conclusion is weak as there is no strong message other than changes occur. It is recommended to introduce some clinical relevance.

 The nature of this study is a proof of concept: we aim to prove that EMG is sensitive enough to detect differences between different DBS adjustments. The results support this hypothesis even though the number of patients in this study does not allow for wider conclusions. We believe that it is essential to have objective means to assess the symptoms of PD and not only for adjustment of DBS. The study shows that the clinical state of the patient may change while the current clinical markers stay unchanged, and it indicates that the clinical assessment is too coarse. The study provides means to assess the changes based on muscle activation patterns. 

 Manuscript updated.

Figures

>>> Figure 1- Please either define the acronyms of the 7 settings in the figure legend or ideally, add as subheadings.

>>> Figure 2 – See above.

 Manuscript updated.

Dear reviewer #2,

Introduction

>>> The introduction should be shortened and should have stronger focus on the need to improve the quality of programming algorithms for DBS therapy.

>>> Description of general medical treatment strategies in PD may be left out as well as hypotheses on the mechanisms of action of DBS.

>>> Instead, a statement that DBS is an appropriate therapeutic option in late stage PD may be enough and may be followed by discussing the problems clinicians and patients are confronted with during conventional DBS programming sessions (try and error, delayed DBS effects and side effects).

#>>> The last paragraph in the introduction may then illustrate why the authors have chosen EMG recordings to objectify clinical DBS effects.

 Manuscript updated. We like to include short description of the general picture of the disease and available treatments before going into details of the study. However, parts related to elementary physiology of PD and mechanisms of DBS have been focused.

Methods

>>> Authors should explain why they have chosen the mentioned motor task. To me, this task is far from being “objective” as the performance of such a task is patient driven in acceleration and speed of the movements and in its muscle strength which directly affects EMG activity.

 The dynamic elbow flexion task or slight variations have been used in multiple PD studies (Rissanen et al 2009, Flament et al 2003, Robichaud et al 2002, Pfann et al 2001). It is true that the task is patient driven in speed of movements. The dynamic movement does not alleviate the symptoms, quite the opposite, it may even provoke the symptoms. As the movement speed in this particular task cannot be controlled precisely without a manipulator, traditional EMG analysis for e.g. amplitude is not helpful. However, we argue that slight variations in movement speed do not signifcantly affect the muscle synchronisation in PD. It has been reported that kurtosis is affected at high speed movement (120 bpm and above), but not lower such as in this study (Ahmad et al. 2009).

 Manuscript updated.

>>> The authors should explain why they have chosen to evaluate such minimal changes of amplitude and frequency compared to the clinically chosen DBS settings, which the authors call “base setup”. I do agree that sometimes subtle changes of DBS parameters may influence motor symptoms of PD but I cannot see any rationale for +-0.3V, +-30Hz, +-30µs.

 In DBS treatment, the effect of the treatment strongly depends on the stimulation settings, voltage, frequency, pulse width. The stimulator has to be tuned individually to determine the optimal stimulation settings for the best treatment effect. These optimal stimulation settings are defined as the base setup in the manuscript. Subtle changes around these optimal values were selected since the purpose of the study was to determine if clinically relevant small steps in tuning DBS can be detected with EMG measurement. The exact steps were selected to reflect typical DBS adjustment session, in which the purpose is to fine tune the DBS.

 Manuscript updated.

>>> I would strongly recommend to additionally study patients in medication / dopamine depleted state as PD motor symptoms may similarly become reduced with medication or DBS. Otherwise, their relative influence on symptom relief cannot be described and therefore effects on EMG recording cannot clearly be assigned to one or the other therapy: What parameters of EMG recordings are influenced by medication? What parameters of EMG recordings are influenced by DBS?

 This is an important point. The medication is typically used along with DBS therapy to achieve optimal symptom control. The patients were studied with their current normal medication as that is the situation when the patients have their stimulator fine tuned. (The first adjustments are done without medication). This was taken into account when planning the study. The measurement duration was kept as low as possible while maintaining enough time for the DBS to stabilise. By doing this had two advantages: the measurement was not burdensome to the patient, but also the medication response was somewhat constant. Further, the different DBS settings (excluding optimal and off) were measured in randomized order to decrease systematic errors such as this. 

 Manuscript updated.

>>> Characteristics of EMG recordings due to side effects (affecting pyramidal tract) should be defined to distinguish “optimal” from “above threshold” stimulation. Especially because the authors describe “side effects” (without further specification of their clinical appearance) due to greater pulse widths.

 Please see explanation below.

 Manuscript updated.

Analysis

>>> The reader may profit from a more detailed clinical / practical view on and explanation of the parameters chosen to be evaluated from EMG recordings: kurtosis, recurrence rate, correlation dimension.

>>> What are the clinical significances of differences in kurtosis, recurrence rate and correlative distribution between PD and healthy controls? What is the physiological meaning of the mentioned parameters? (onko tätä kysymystä siirretty?)

 Manuscript updated.

Results

>>> Table 1 shows that “base setup” may not be “best setup” as the increase of the parameters amplitude, frequency or pulse width, even in such narrow margins (see above), may further improve DBS clinical efficacy. Authors should discuss and may further analyse differences in EMG recordings which may distinguish patients with suboptimal stimulation from optimal stimulation (e.g. complete clinical rigidity control as clinical feeback).

 This is a good point. We should emphasize that even though some changes were seen, the statistical tests (Wilcoxon) show non significant change in tremor and rigidity scores of the patients during the measurements. In other words, changes in UPDRS-III rigidity and tremor subscore did not show difference between the setups. Full UPDRS-III motor assessment was done with A0 and with DBS off if possible. Full UPDRS-III changed significantly between these two phases.

 Manuscript updated.

>>> Figure 2 shows results of EMG-parameters in relation to “base setup” (A0). Although clinically more effective to weaken tremor and rigidity (see table 1), A+ is still greater 1, suggesting A+ to be “worse” than A0. The authors may explain 1) why they have chosen to analyse EMG parameters relative to A0 and 2) why A+ performs less effective than A0 in the EMG parameters although clinically better in the reduction of PD motor symptoms (same for F+ which performs worse than DBS OFF in the EMG parameters, although of better clinical efficacy than A0).

 1) In the relative analysis, each patient is compared to their own optimal settings (arms separately). The EMG parameters were analysed in relation to A0 to be able to compare the change between the parameters, not absolute values. The comparison could be relative to any other parameter, possibly to DBS off. We do believe that it is most useful to compare the parameter values to A0 since it was the starting point and also the optimal setting for the patients.

 2) There was no significant differences in clinical rigidity and tremor subscores. However, this is a very complicated question. The ultimate goal of adjustment of DBS is to improve patient's motor function and eventually the life quality. There are multiple factors that affect the goodness of DBS adjustment of which arm rigidity and tremor are only a part of. It is possible that patients clinical state seems to be improved based on motor assessment, but the patient has adverse effects or just "bad sensation". Ultimately it is the patient together with neurologist who judge between the adjustments and the motor evaluation is just a tool to aid the decision. This study focuses on changes in EMG morphology and thus motor performance of the patient. The methods presented in this study do not allow for "comprehensive assessment of DBS adjumstent" since there are other motor (posture, gait) and non-motor (side effects: dysartria, diplopia, possibly even gambling?) factors that affect the optimal adjustment.

 The patients may have multiple optimal settings for different type daily activities and sometimes the goal of adjustment is to increase the therapeutic range of the stimulator. In this case the goal is to adjust the stimulator in a way that the clinical parameters do not change.

 Manuscript updated.

>>> The authors should explain and name the mentioned “side effects” of DBS due to changed parameters (e.g. pulse width). DBS may have affected the fibres of the pyramidal tract? What are the effects of these “side effects” on EMG recordings? Again, authors may analyse / discuss how to distinguish suboptimal, optimal and above threshold stimulation (side effects) by means of EMG recordings.

 Manuscript updated.

 There was substantial overlap between previous comments regarding side effects, suboptimal stimulation and we have refactored them to three new comments: 

>>> The authors should explain and name the mentioned “side effects” of DBS due to changed parameters (e.g. pulse width). Especially because the authors describe “side effects” (without further specification of their clinical appearance) due to greater pulse widths. What are the effects of these “side effects” on EMG recordings? DBS may have affected the fibres of the pyramidal tract? 

 Patients were carefully observed for side effects, and they were also advised to immediately report subjective changes. According to our research protocol, original DBS settings (base setup) were immediately restored when clear side effects appeared. Hence, we were not able to record reliable EMG signal during side effects.

 Side effects were observed in total of 17 measurement phases. Most side effects were caused by increasing pulse width. Six patients developed dysarthria probably by unwanted stimulation of corticobulbar fibers. Three patients had muscle contraction probably due to stimulation of corticospinal fibers. One patient developed diplopia due to stimulation of oculomotor nerve. Seven patients developed dyskinesia due to stimulation. All side effects vanished when original DBS settings were restored.

 Generally rapid limb movements or muscle contraction may be detected from EMG as signal amplitude changes. Mild twitching and tingling may not be visible, but can be observed as change of signal morphology. Non-motor symptoms e.g. dysartria, diplopia cannot be directly measured with EMG. 

 Manuscript updated.

>>> Authors should discuss and may further analyse differences in EMG recordings which may distinguish patients with suboptimal stimulation from optimal stimulation (e.g. complete clinical rigidity control as clinical feeback). Again, authors may analyse / discuss how to distinguish suboptimal, optimal and above threshold stimulation (side effects) by means of EMG recordings. Characteristics of EMG recordings due to side effects (affecting pyramidal tract) should be defined to distinguish “optimal” from “above threshold” stimulation.

 This is an interesting question. In this study, the patients reacted individually to non-optimal settings. Some patients experienced dystonic movements while others had dysarthria. No generalisations about how or which side effects different stimulation settings cause can be made based on the results.

 Non-optimal stimulation settings cause changes to EMG signal morphology. With optimal settings DBS regulates the motor control system in a way that pathological synchronisation decreases. Thus, sub-threshold and above threshold stimulation may have similar effects to EMG signal. 

 Manuscript updated.

>>> Statistical results are not rigorously stated (performed tests and resulting values are missing).

 Manuscript updated.

Discussion

>>> The “U-shaped theory” of optimal stimulation parameters needs to be better explained. “EMG parameters had their extremum at the base setup” is not shown in the results section (only relative values in fig2).

 Unfortunately we do not understand the comment. Montgomery et al. have suggested suggested that increase in stimulation voltage improves the symptoms only until a certain point is reached. After this point the symptoms get worse if voltage is further increased. This theory has been used as an analogy for interpreting the results of this study.

 Manuscript updated.

>>> The whole manuscript needs major editing concerning language and spelling.

 Manuscript updated.

---

## [Decision Letter · Decision Letter 1]

23 Dec 2021

PONE-D-21-16366R1Elbow flexion EMG morphology changes during adjustment of deep brain stimulator in advanced Parkinson's diseasePLOS ONE

Dear Dr. Ruonala,

Thank you for submitting your manuscript to PLOS ONE. After careful consideration, we feel that it has merit but does not fully meet PLOS ONE’s publication criteria as it currently stands. Therefore, we invite you to submit a revised version of the manuscript that addresses the points raised during the review process.

We look forward to receiving your revised manuscript.

Kind regards,

Karsten Witt

Academic Editor

PLOS ONE

Reviewers' comments:

Reviewer's Responses to Questions

**Comments to the Author**

1. If the authors have adequately addressed your comments raised in a previous round of review and you feel that this manuscript is now acceptable for publication, you may indicate that here to bypass the “Comments to the Author” section, enter your conflict of interest statement in the “Confidential to Editor” section, and submit your "Accept" recommendation.

Reviewer #2: (No Response)

Reviewer #3: (No Response)

2. Is the manuscript technically sound, and do the data support the conclusions?

Reviewer #2: Yes

Reviewer #3: Partly

3. Has the statistical analysis been performed appropriately and rigorously? 

Reviewer #2: Yes

Reviewer #3: No

4. Have the authors made all data underlying the findings in their manuscript fully available?

Reviewer #2: No

Reviewer #3: No

5. Is the manuscript presented in an intelligible fashion and written in standard English?

Reviewer #2: No

Reviewer #3: No

6. Review Comments to the Author

Reviewer #2: The authors present a re-structured and re-written manuscript of their work aiming to correlate DBS-effects with EMG-recordings. To me, the manuscript has much improved and almost all of my concerns were satisfactorily answered.

Nevertheless, there are still few concerns:

- The Abstract ends with the sentence “The parameters had their extremum at optimal clinical settings” which is (without having studied the whole manuscript) contextually not understandable and thus needs to be further explained.

- Introduction ll15-17: The authors may leave out these useless phrases concerning DBS mechanisms of action (as their work does not deal with this topic at all) or have to explain the current opinions of DBS mechanisms of action further in detail.

- Discussion ll 226-232: I would interpret this U-shape theory that symptoms may get worse when voltage is further increased by side effects due to current spread to neighbouring structures rather than due to worsening of PD symptoms per se. The latter may become worse when voltage is decreased. Accordingly, authors should also revise (or further explain) the phrase “Adjusting the DBS further, increasing or decreasing voltage of frequency or increasing pulse width, caused the parameters to get worse as more parkinsonian features get to the signal”.

- Reference 33 is obviously not correct.

- The whole manuscript may still profit from a strict proof reading according to spelling and language, the latter preferably by an english native speaker.

Reviewer #3: In their paper Ruona explored the effect of changes in DBS setting on EMG signals . The rationale of the study is well received since optimising DBS parameters is still an empirical, time consuming and rather subjective process. The work build on earlier findings in which the group now aimed to see whether optimal settings could be differentiated from sub-optimal settings. Their key findings were that there were significant differences between the EMG characteristics but not between clinical scores measured with UPDRS scores.

Although the findings are interesting, many answers are still missing. For example in how much % of the cases a combination of the EMG characteristics can predict the optimal settings. This would be more informative. Furthermore, it is very well possible that small changes in DBS settings don’t elicit lead to EMG changes. For this reason, the authors could make use of only those settings that resulted in a different clinical score and perform ROC analyses.

Were wash-out of DBS times taken into account?

Was the inter-rater agreement of the UPDRS scores known?

minor

> abstract > I’m missing numbers in the abstract, significant should be mentioned

> quantify between settings? > this is prob rather difficult, isn’t it easier to differentiate between effective and non-effective parameters?

> intro > more than a decade

7. PLOS authors have the option to publish the peer review history of their article (what does this mean?). If published, this will include your full peer review and any attached files.

Reviewer #2: No

Reviewer #3: No

---

## [Author Response · Author response to Decision Letter 1]

24 Jan 2022

1. If the authors have adequately addressed your comments raised in a previous round of review and you feel that this manuscript is now acceptable for publication, you may indicate that here to bypass the “Comments to the Author” section, enter your conflict of interest statement in the “Confidential to Editor” section, and submit your "Accept" recommendation.

 Reviewer #2: (No Response)

 Reviewer #3: (No Response)

2. Is the manuscript technically sound, and do the data support the conclusions?

 Reviewer #2: Yes

 Reviewer #3: Partly

3. Has the statistical analysis been performed appropriately and rigorously?

 Reviewer #2: Yes

 Reviewer #3: No

4. Have the authors made all data underlying the findings in their manuscript fully available?

 Reviewer #2: No

 Reviewer #3: No

 We would like to emphasise that we have already provided data underlying the findings described in the manuscript (supplementary_information.xlsx). Unfortunately we are unable to share the raw patient data related to the manuscript due to local regulations and the ethics statement related to the project.

5. Is the manuscript presented in an intelligible fashion and written in standard English?

 Reviewer #2: No

 Reviewer #3: No

6. Review Comments to the Author

Reviewer #2: 

 The authors present a re-structured and re-written manuscript of their work aiming to correlate DBS-effects with EMG-recordings. To me, the manuscript has much improved and almost all of my concerns were satisfactorily answered.

 Nevertheless, there are still few concerns:

 - The Abstract ends with the sentence “The parameters had their extremum at optimal clinical settings” which is (without having studied the whole manuscript) contextually not understandable and thus needs to be further explained.

 > Manuscipt updated to clarify this.

 - Introduction ll15-17: The authors may leave out these useless phrases concerning DBS mechanisms of action (as their work does not deal with this topic at all) or have to explain the current opinions of DBS mechanisms of action further in detail.

 > Manuscript updated.

 - Discussion ll 226-232: I would interpret this U-shape theory that symptoms may get worse when voltage is further increased by side effects due to current spread to neighbouring structures rather than due to worsening of PD symptoms per se. The latter may become worse when voltage is decreased. Accordingly, authors should also revise (or further explain) the phrase “Adjusting the DBS further, increasing or decreasing voltage of frequency or increasing pulse width, caused the parameters to get worse as more parkinsonian features get to the signal”.

 > Yes, you're right. U-shaped theory concerns "motor performance" and may be interpreted as combination of symptoms and side effects. 

 >Manuscript updated.

 - Reference 33 is obviously not correct.

 > Manuscript updated, reference omitted - recurrence quantification analysis is a well established methodology. Detailed references are provided later on the manuscript.

 - The whole manuscript may still profit from a strict proof reading according to spelling and language, the latter preferably by an english native speaker.

 > Manuscript has been proofread by an english native speaker. The corrections of spelling and language are not indicated in the revised manuscript.

Reviewer #3: 

 In their paper Ruona explored the effect of changes in DBS setting on EMG signals. The rationale of the study is well received since optimising DBS parameters is still an empirical, time consuming and rather subjective process. The work build on earlier findings in which the group now aimed to see whether optimal settings could be differentiated from sub-optimal settings. Their key findings were that there were significant differences between the EMG characteristics but not between clinical scores measured with UPDRS scores.

 - Although the findings are interesting, many answers are still missing. For example in how much % of the cases a combination of the EMG characteristics can predict the optimal settings. This would be more informative. 

 > This is an interesting suggestion. We want to emphasise, that the nature of the study is more like proof of concept. While the results show that in most of the cases the parameters indicate the optimal amplitude, frequency or pulse width, the number of patients is still low for statistics. Figure 2 has been updated to clarify the differences between the patients. Manuscript has been updated.

 - Furthermore, it is very well possible that small changes in DBS settings don’t elicit lead to EMG changes. For this reason, the authors could make use of only those settings that resulted in a different clinical score and perform ROC analyses.

 >Contrary to our expectations, there was only small change in clinical parameters (arm tremor and rigidity) during the adjustment, not even turning the stimulator off caused significant increase in these variables. (There was a significant increase in full UPDRS-III assessment though.) Unfortunately the data (N=13) does not allow for further splitting. However we believe that ROC analysis would be an interesting to test in further studies with larger data set.

 >Manuscript has been updated to address this.

 - Were wash-out of DBS times taken into account?

 > The patient's state was let to stabilise minimum of five minutes after the adjustment of DBS before beginning the measurement. We believe that it is enough for rapidly relieving symptoms (rigidity, tremor). Other symptoms may take longer time to stabilise and that is an inherent challenge - the observation time should be much longer. We tried to balance between long enough time for adequate stabilisation while at the same time keeping the total duration of measurements sufficiently short, 2,5 hours.

 - Was the inter-rater agreement of the UPDRS scores known?

 > Inter-rater agreement was not evaluated in this study. All UPDRS evaluations were made by same experienced neurologist specialised to Parkinson's disease.

 minor

 - abstract > I’m missing numbers in the abstract, significant should be mentioned

 > Manuscript updated: Significant results related to UPDRS and significant changes in recurrence rate (table 2) were included in the abstract.

 - quantify between settings? > this is prob rather difficult, isn’t it easier to differentiate between effective and non-effective parameters?

 > This is an excellent comment. Manuscript has been updated to emphasise that comparisons were only made between the optimal setup and other setups.

 - intro > more than a decade

 > Manuscript updated.

7. PLOS authors have the option to publish the peer review history of their article (what does this mean?). If published, this will include your full peer review and any attached files.

Do you want your identity to be public for this peer review? For information about this choice, including consent withdrawal, please see our Privacy Policy.

 Reviewer #2: No

 Reviewer #3: No

---

## [Decision Letter · Decision Letter 2]

31 Mar 2022

Changes in elbow flexion EMG morphology during adjustment of deep brain stimulator in advanced Parkinson's disease

PONE-D-21-16366R2

Dear Dr. Ruonala,

We’re pleased to inform you that your manuscript has been judged scientifically suitable for publication and will be formally accepted for publication once it meets all outstanding technical requirements.

Kind regards,

Karsten Witt

Academic Editor

PLOS ONE

---

## [Editor Report · Acceptance letter]

7 Apr 2022

PONE-D-21-16366R2 

Changes in elbow flexion EMG morphology during adjustment of deep brain stimulator in advanced Parkinson's disease 

Dear Dr. Ruonala:

I'm pleased to inform you that your manuscript has been deemed suitable for publication in PLOS ONE. Congratulations! Your manuscript is now with our production department. 

Kind regards, 

on behalf of

Dr. Karsten Witt 

Academic Editor

PLOS ONE